# *Gm14230* controls Tbc1d24 cytoophidia and neuronal cellular juvenescence

**Takao Morimune**[1,2,3], **Ayami Tano**[1,3], **Yuya Tanaka**[1,3], **Haruka Yukiue**[1], **Takefumi Yamamoto**[4], **Ikuo Tooyama**[1], **Yoshihiro Maruo**[2], **Masaki Nishimura**[1], **Masaki Mori**[1,3]*

1 Molecular Neuroscience Research Center (MNRC), Shiga University of Medical Science, Seta Tsukinowa-cho, Otsu, Shiga, Japan, 2 Department of Pediatrics, Shiga University of Medical Science, Seta Tsukinowa-cho, Otsu, Shiga, Japan, 3 Department of Vascular Physiology, National Cerebral and Cardiovascular Center Research Institute, Suita, Osaka, Japan, 4 Central Research Laboratory, Shiga University of Medical Science, Seta Tsukinowa-cho, Otsu, Shiga, Japan

* morim@belle.shiga-med.ac.jp, mori.masaki@ncvc.go.jp

**Data Availability Statement:** All relevant data are within the manuscript and its Supporting Information files.

**Funding:** T.M. is supported by a research grant from the Morinaga Service Society. M.M. is

## Abstract

It is not fully understood how enzymes are regulated in the tiny reaction field of a cell. Several enzymatic proteins form cytoophidia, a cellular macrostructure to titrate enzymatic activities. Here, we show that the epileptic encephalopathy-associated protein Tbc1d24 forms cytoophidia in neuronal cells both *in vitro* and *in vivo*. The Tbc1d24 cytoophidia are distinct from previously reported cytoophidia consisting of inosine monophosphate dehydrogenase (Impdh) or cytidine-5'-triphosphate synthase (Ctps). Tbc1d24 cytoophidia is induced by loss of cellular juvenescence caused by depletion of *Gm14230*, a juvenility-associated lncRNA (JALNC) and zeocin treatment. Cytoophidia formation is associated with impaired enzymatic activity of Tbc1d24. Thus, our findings reveal the property of Tbc1d24 to form cytoophidia to maintain neuronal cellular juvenescence.

## Introduction

We previously identified juvenility-associated genes (JAGs) that were selectively highly expressed in juvenile tissues [1]. By extending our analysis to the noncoding element of the transcriptome, we further identified juvenility-associated long noncoding RNAs (JALNCs) as lncRNAs that were predominantly expressed in juvenile [2]. Among the JALNCs, *Gm14230* maintains cellular juvenescence that is characterized by cellular properties of growth, differentiation capacity and resistance to cellular senescence [2]. It is not completely understood how *Gm14230* depletion leads to the loss of cellular juvenescence. Here, we show that *Gm14230* depletion causes formation of a cellular macrostructure called cytoophidia consisting of Tbc1d24 protein.

*Tbc1d24* is expressed abundantly in the cerebral cortex in juvenile mice and encodes a GTPase activating protein (GAP) that regulates cytoskeletal reorganization, vesicular transportation and cellular migration [3]. Tbc1d24 possesses functional domains such as a TBC (Tre2-Bub2-Cdc16) domain with Rab GAP activity [4, 5] and a TLDc (domain catalytic) domain that mediates resistance to oxidative stress [6, 7]. Tbc1d24 GAP regulates small GTPases Arf6 [3]

supported by research grants from the Kato Memorial Bioscience Foundation, the Japan Epilepsy Research Foundation (JERF), the Hoansha Foundation, MSD Life Science Foundation, the Ichiro Kanehara Foundation for the Promotion of Medical Sciences and Medical Care, Japan Brain Foundation, Takeda Science Foundation and the Japan Spina Bifida & Hydrocephalus Research Foundation. This study was supported by Grants-in-Aid for Scientific Research for Young Scientists from the Japan Intractable Diseases (Nanbyo) Research Foundation. This study was supported by JSPS KAKENHI Grant Numbers 15H01486, 18K07788 and 19H04774, the Leading Initiative for Excellent Young Researchers (LEADER) 5013323 and the Initiative on Rare and Undiagnosed Diseases (IRUD) of AMED. The funders had no role in study design, data collection and analysis, decision to publish, or preparation of the manuscript.

**Competing interests:** The authors have declared that no competing interests exist.

and Rab35 [8]. Arf6 is implicated in membrane trafficking between plasma membrane and endocytic compartments through activation of lipid-modifying enzymes phospholipase D and phosphatydylinositol-4-phosphate 5 kinase (PIP5K) [9, 10]. Rab35 mediates membrane trafficking in synapses and between plasma membrane and early endosomes [11]. Rab35 and Arf6 antagonize each other in membrane trafficking [12]. Tbc1d24 interacts with EphrinB2 and plays a role in locomotion of neural crest cells [8]. However, it remains to be completely understood how the enzymatic function of Tbc1d24 is regulated in a cell.

Mutations in *TBC1D24* cause the refractory seizure syndrome called epileptic encephalopathy (EE). *TBC1D24* mutations have been reported in patients with DOORS (deafness onychodystrophy, osteodystrophy, mental retardation and seizures [13]) syndrome, progressive myoclonic epilepsy [14], early infantile EE [15] and malignant migrating partial seizures of infancy (MMPSI, [16]). A truncating mutation of *TBC1D24* results in severe neurodegeneration [17]. Consistently, *Tbc1d24* knockout mice displayed spontaneous seizures and abnormal behaviors with compromised axonal growth [18], vesicle trafficking [19, 20] and dendritic spine formation in cortical and hippocampal neurons [21].

The cytoophidium is a macromolecular ring- or rod-like structure found in the cytoplasm [22–24]. Cytoophidia have been reported to be formed by polymerization of the enzymatic proteins cytidine-5'-triphosphate synthase (Ctps, [25–29]), inosine monophosphate dehydrogenase (Impdh, [30–36]), asparagine synthetase [37] and phosphofructokinase-1 (Pfk1, [38]).

We here show that Tbc1d24 forms cytoophidia in neuronal cells both *in vitro* and *in vivo*. Tbc1d24 cytoophidia were induced by loss of cellular juvenescence caused by *Gm14230* depletion and zeocin-induced genotoxicity. Tbc1d24 cytoophidia were distinct from the previously characterized cytoophidia formed by Impdh and Ctps. The GAP activity of Tbc1d24 is suppressed by forming cytoophidia. The loss of cellular juvenescence enhances formation of the cytoophidia that has a protective role against the cellular stress. Thus, Tbc1d24 cytoophidia underlie the physiological functions of juvenile neuronal cells.

## Materials and methods

### Mouse experiment

All animal experiments were approved by the institutional animal care and use committee at Shiga University of Medical Science. All experiments were performed in accordance with the relevant guidelines and regulations. The brains of C57BL/6N mice perfused with 4% paraformaldehyde (PFA) in phosphate-buffered saline (PBS) on the postnatal day 11 (P11) were dissected and post-fixed with 4% PFA in PBS. The brains were cryoprotected in 15% sucrose in PBS for 3 days at 4°C. The brains were embedded in optimum cutting temperature (OCT) compound (Sakura Finetek, Tokyo, Japan), frozen and stored at -80°C until use. Brains were sectioned at 20 μm thickness using a cryostat (CM3050, Leica) and mounted onto MAS-coated slide glass (Matsunami glass, S9441, Osaka, Japan). The sections were washed 3 times in 0.1% Triton X-100 in PBS (PBS-T), and nonspecific staining was blocked by treatment with 3% bovine serum albumin (BSA) in PBS-T at room temperature (RT) for 1 hour (hr). The sections were incubated with primary antibodies overnight at 4°C, washed 3 times in PBS-T, and then incubated with secondary antibodies overnight at 4°C. The information for the antibodies were shown below. After washing 3 times in PBS-T, the slides were mounted with Prolong Gold Antifade mountant with DAPI (Invitrogen).

### Cell culture

Neuro2a and SH-SY5Y cells were cultured in Eagle's minimal essential medium supplemented with 10% fetal bovine serum (FBS) and Ham's F12:EMEM (1:1) supplemented with 10% FBS,

respectively. To model loss of cellular juvenescence, 200 μg/ml zeocin was added to the culture medium for 72 hrs or 96 hrs as indicated in Figure legend. For detection of dead cells, Sytox blue dye (Thermo Fischer, S11348, Massachusetts, US) was added to the culture medium at 5 μM. For evaluation of Impdh or Ctps cytoophidia, Acivicin or 6-diazo-5-oxo-L-norleucine (DON) was added to the culture medium at 2 mM for 24 hrs. For induction of Impdh cytoophidia, cells were treated with 2 μM mycophenolate (MPA) for 24 hrs. Cell images were obtained with EVOS FL cell imaging system (Thermo Fisher Scientific). The number of cells per field was counted with Fiji software (https://imagej.net/Fiji, [39]).

## Plasmid construction

The *Gm14230* sequence was amplified by PCR with the primers (Forward: 5′-GTTTCCGGA GCGCCTACT-3′, Reverse: 5′-CTGTTCTGAGTCGCTCTTGC-3′) and a cDNA library derived from mouse NIH3T3 cells. The PCR product was first cloned into the pMD20-T vector (Takara BIO, Shiga, Japan) for sequence confirmation. The cloned product was then subcloned into pc.DNA-3.1(-) using *HindIII* and *BamHI* sites.

## Transfection in cultured cells

For knockdown experiments, Lipofectamine RNAiMAX (Invitrogen, California, US) was used to transfect siRNA duplexes at 20 nM: mouse *Tbc1d24* siRNA1 (Sigma-Aldrich, Mission siRNA SASI_Mm01_00145836, Missouri, US), *Tbc1d24* siRNA2 (Sigma-Aldrich, Mission siRNA SASI_Mm01_00145838), mouse *Gm14230* siRNA1 (5′-GGUGCUAAAGGACCAGUUG dTdT-3′), G*m14230* siRNA2 (5′-GCUCAGUCGGCUCAGAGUAdTdT-3′) and siRNA negative control (Invitrogen, AM4611). The knockdown efficiency was assessed 48 hrs after transfection by qPCR or western blot analysis.

In overexpression experiments, the *Gm14230* plasmid was transfected into Neuro2a cells with Lipofectamine 2000 (Invitrogen). The extent of forced expression was assessed 48 hrs after transfection by qPCR.

## Real-time quantitative PCR

Total RNA was extracted using TRIzol reagent (Thermo Fisher Scientific). The extracted RNA was quantified using a NanoDrop Lite Spectrophotometer (Thermo Fisher Scientific) and reverse transcribed using a high-capacity RNA to cDNA kit (Applied Biosystems, California, US) and PCR thermal cycler Dice (Takara BIO) according to the manufacturer's instructions. qPCR was performed using a LightCycler 480 SYBR Green I Master Kit on a LightCycler 480 instrument (Roche, Basel, Switzerland) using reverse transcribed cDNA as a template. The specificity and quality of qPCR amplification was assessed by agarose gel electrophoresis and a melting curve analysis. The expression levels were normalized to mouse *Tubb5* or *Polr2a* as indicated in the Figure legends. The primers used for qPCR are mouse *Tbc1d24* (Forward: 5′-AGCACTGAGGCAGAAGGGTA-3′, Reverse: 5′-CATCTCCTTCACGCTGACAA-3′), *Gm14230* (Forward: 5′-CTTACCACGTGTGCCAGTGT-3′, Reverse: 5′-CTGGGGTCAC TGGTGGAAT-3′), *Tubb5* (Forward: 5′-GATCGGTGCTAAGTTCTGGGA-3′, Reverse: 5′- AGGGACATACTTFCCACCTGT-3′) and *Polr2a* (Forward: 5′-GAGTCCAGAACGAGTGCA TGA-3′, Reverse: 5′-ACAGGCAACACTGTGACAATC-3′).

## Western blot analysis

Cell lysates were prepared using RIPA lysis buffer containing 25 mM Tris-HCl (pH 7.6), 150 mM NaCl, 1% NP-40, 1% sodium deoxycholate and 0.1% SDS. The lysates were centrifuged at

15,000 rpm for 10 min. The lysates were mixed with Laemmli sample buffer and boiled at 98˚C for 2 min. The protein samples were run on SDS-PAGE along with protein markers for SDS-PAGE (Nacalai Tesque, 02525–35, Kyoto, Japan) and blotted onto a PVDF membrane (GE healthcare, Illinois, US). The following primary antibodies were used: Tbc1d24 (Protein-Tech, 1:1000, Illinois, US; LSBio, 1:1000, Illinois, US), p21 Waf1 (Abcam, ab109199, 1:1000, Cambridge, UK), p27 Kip1 (Cell Signaling Technology, #3698, 1:1000, Massachusetts, US), Impdh (ProteinTech, 12948-1-AP, 1:100), Cpts1 (ProteinTech, 15914-1-AP, 1:100), Arf6 (Cell Biolabs, STA-407-6, 1:500, California, USA) and beta-actin (Cell Signaling Technology, #5174, 1:4000). HRP-conjugated anti-mouse IgG (Thermo Fisher Scientific, 32430) and anti-rabbit IgG (Thermo Fisher Scientific, 32460) were used as secondary antibodies at 1:1000 dilution. For detection of immunoreactive bands, Chemi-Lumi One L or Chemi-Lumi One Ultra (Nacalai Tesque) were used. Uncropped blots were shown in **S1 Raw images**. Densitometric analyses for blotting data were performed using Fiji [39].

### Arf6 activity assay

The active form of Arf6 was detected using a GTP-bound Arf6-specific pulldown assay kit (Cell biolabs, STA-407-6). Cells were lysed in Arf6 lysis buffer containing 25 mM HEPES, pH 7.5, 150 mM NaCl, 1% NP-40, 10 mM MgCl2, 0.2 mM EDTA and 2% glycerol, followed by centrifugation at 15,000 rpm for 10 min at 4˚C. The supernatant was incubated with Golgi-localized γ-ear containing Arf-binding protein 3 protein binding domain (GGA3 PBD) agarose beads for 1 hr at 4˚C. Beads were washed 3 times in Arf lysis buffer. The pellets were added to 2x Laemmli sample buffer and boiled at 95˚C for 5 min. The supernatants were analyzed with SDS-PAGE and western blot analysis with Arf6 antibody (Cell Biolabs, STA-407-6, 1:500). HRP-conjugated anti-mouse IgG (Thermo Fisher Scientific, 32430) was used as the secondary antibody at 1:1000 dilution.

### Cycloheximide chase assay

To access whether cytoophidia formation affected the half-life of Tbc1d24 protein, Tbc1d24 protein expression levels were analyzed in Neuro2a cells treated with zeocin at 200 μg/ml or control distilled water for 72 hrs in the presence of cycloheximide (CHX) at 20 μg/ml. Cell lysates were prepared at 0, 4, 8, 12 hrs after the CHX treatment and analyzed for the expression of Tbc1d24 protein by western blots as described above.

### Co-immunoprecipitation assay

To analyze potential interactions of Tbc1d24 with Ctps and Impdh, Neuro2a cells were transfected with FLAG tagged *TBC1D24* plasmid (GenScript). Forty-eight hrs after transfection, cells were lysed with NETN buffer containing 20 mM Tris-HCl (pH 8.0), EDTA 1 mM, NaCl 100 mM and 0.5% NP-40, followed by centrifugation at 20,800x g for 15 min at 4˚C to remove debris. Protein G Sepharose beads (Invitrogen) were incubated with 2 μg anti-FLAG antibody or control rabbit IgG (Cell Signaling Technology, #2729) for 1 hr at RT, and then incubated with the cell lysates overnight at 4˚C. The beads were then washed with NETN200 wash buffer containing 20 mM Tris-HCl, 1 mM EDTA, 200 mM NaCl and 0.5% NP-40 four times, followed by elution with Laemmli sample buffer and denaturation at 95˚C for 5 min. The samples were analyzed by western blots with Impdh, Ctps and FLAG antibodies.

## Senescence-associated beta-galactosidase activity

For staining of senescence-associated beta-galactosidase (SA-β-gal) activity, Neuro2a cells transfected with *Gm14230* siRNA were cultured for 7 days followed by fixation with 3% (v/v) formaldehyde. The cells were then washed with PBS twice and incubated with the staining solution containing 5 mM $K_3Fe(CN)_6$, 2 mM $MgCl_2$, 150 mM NaCl, 30 mM citric acid/phosphate buffer, 5 mM $K_4Fe(CN)_6$, and 1 mg/ml X-Gal for 16 hrs at 37°C.

## Immunocytochemistry

Cells were fixed with 4% PFA at RT for 10 min and permeabilized with 0.1% Triton X-100 in PBS for 2 min. For blocking, cells were incubated with 2% FBS in PBS for 1 hr at RT. Cells were then incubated at 4°C overnight with the following primary antibodies: Tbc1d24 (ProteinTech, 25254-1-AP, 1:200, LSBio, LS-C679739, 1:200, Washington, US), Vimentin (Santa Cruz, sc-6260, 1:100, Texas, US), Impdh (Santa Cruz, sc-36-5171, 1:100, ProteinTech, 12948-1-AP, 1:100), Cpts1 (ProteinTech, 15914-1-AP, 1:200 and LSBio, LS-C197001, 1:100). After washing with PBS, cells were incubated with anti-mouse IgG conjugated with Alexa Fluor 546 (Invitrogen, A11030, 1:1000) or anti-rabbit IgG conjugated with Alexa Fluor 488 (Invitrogen, A11008, 1:1000) for 1 hr at RT. Prolong Gold Antifade mountant with DAPI (Invitrogen) was used for mounting and nuclear staining. Cellular images were taken with a confocal microscope (Leica, TCP SP8, Wetzlar, Germany) or a super-resolution fluorescence microscope (GE healthcare, DeltaVision Elite). The length (μm) of cytoophidia was measured using Fiji [39].

## Analyses of cytoophidia in various conditions

To assess the Acivicin-responsibility of Tbc1d24 cytoophidia, Neuro2a cells were treated with 2mM Acivicin or control distilled water for 24 hrs. Acivicin is a known inducer of Impdh and Ctps cytoophidia. Immunofluorescence analyses for Tbc1d24, Impdh and Ctps were conducted to assess the influences on cytoophidia formation.

To assess reversibility of Tbc1d24 cytoophidia formation, Neuro2a cells were treated with 2 μM MPA or dimethyl sulfoxide (DMSO) for 24 hrs before washing out MPA. The cells were then further cultured for 24 or 48 hrs and analyzed for Tbc1d24 cytoophidia by immunocytochemistry described as above.

For the assessment of a protective effect of Tbc1d24 in loss of cellular juvenescence, Neuro2a cells were transfected simultaneously with *Gm14230* (or control) siRNA and pcDNA3.1--FLAG-*TBC1D24* (or empty) plasmid. Cell growth were analyzed 48 hrs after the transfection.

To assess the effect of forced expression of *Tbc1d24* in zeocin-induced cellular juvenescence loss, Neuro2a cells were transfected with pcDNA3.1-FLAG-*TBC1D24* plasmid or control empty pcDNA3.1 plasmid. Twenty-four hrs later, the cells were treated with zeocin at 200 μg/ml or distilled water for 72 hrs for the assessment of cell growth.

To assess a role of Tbc1d24 in loss of cellular juvenescence induced by *Gm14230* depletion, Neuro2a cells were transfected simultaneously with *Tbc1d24* and *Gm14230* siRNAs. The cell growth and viability were analyzed 48 hrs after the transfection. For the viability assessment, Sytox blue staining was performed described as above.

## Gene expression analysis in mouse cerebral cortexes

*Tbc1d24* expression in mouse cerebral cortexes at P1 and P56 was analyzed by the strand-specific RNA-seq dataset we previously deposited with the accession number DRA009510. The RNA-seq data were processed using BioLinux8 [40]. The expression was visualized using Integrative Genomic Viewer (http://software.broadinstitute.org/software/igv/). Gene expression

levels show fragments per kilobase of transcript per million fragments sequenced (FPKM) at P1 and P56.

## Statistical analysis

For all quantified data, the mean ± standard error of the mean (SEM) was presented. Statistical significance between two experimental groups was indicated by an asterisk, and comparisons were made using Student's *t*-test. *P*-values less than 0.05 were considered significant.

## Results

### Tbc1d24 forms cytoophidia

We previously performed a comprehensive transcriptomic analysis to elucidate mechanisms for the juvenile properties of animals and identified JAGs and JALNCs. Among the JAGs, we focused on *Tbc1d24*, the gene predominantly highly expressed in juvenile compared to adult in the cerebral cortex (**Fig 1A, S1 Table**). We validated the juvenile-predominant expression of *Tbc1d24* by real-time qPCR (**Fig 1B**). We performed immunostaining of Tbc1d24 in Neuro2a mouse neuroblastoma cells and found that Tbc1d24 took a ring-like macrostructure in the cytoplasm (**Fig 1C**). Similar macrostructures, called as cytoophidia, have been reported with proteins such as Impdh and Ctps. Super-resolution imaging further revealed Tbc1d24 took a string-like structure (**Fig 1D**). The Tbc1d24 cytoophidia was also detected with the other antibody developed against the distinct region of Tbc1d24 protein (**Fig 1E**). To examine whether Tbc1d24 cytoophidia was formed in the mouse cerebral cortex, we performed immunohistochemistry for Tbc1d24 and observed the similar string structure in the cerebral cortex of mice (**Fig 1F**). Tbc1d24 cytoophidia was observed also in SH-SY5Y, human neuroblastoma cells (**Fig 1G**), indicating Tbc1d24 cytoophidia was conserved across species.

### Tbc1d24 cytoophidia is distinct from Impdh and Ctps cytoophidia

To examine whether Tbc1d24 cytoophidia was distinct from previously known structures, we investigated a contribution of proteins that have been reported to form cytoophidia. The expression levels of the cytoophidia-related genes *Impdh* and *Ctps* were also higher in juvenile than in adult (**S1 Table**). Immunostaining with Impdh (**Fig 2A**) or Ctps (**Fig 2B**) did not show any colocalization to Tbc1d24 cytoophidia. Co-immunoprecipitation assay with Neuro2a cells expressing FLAG-*Tbc1d24* did not show the appreciable interactions of Tbc1d24 protein with Impdh and Ctps protein (**S1 Fig**). The distinction of Tbc1d24 cytoophidia from Impdh and Ctps cytoophidia were further assessed as follows. Impdh cytoophidia is known to be induced by 6-diazo-5-oxo-L-norleucine (DON), Acivicin [41] and mycophenolate (MPA). DON and Acivicin are glutamine analogs and inhibit glutamine-dependent enzymes [42]. Both are strong inducer of Impdh and Ctps cytoophidia [22, 43]. MPA reversibly inhibits Impdh activity that converts IMP to guanosine monophosphate (GMP) [44]. MPA induces Impdh cytoophidia [43]. Tbc1d24 cytoophidia were decreased significantly by DON (**Fig 2C** and **2D**) and MPA (**Fig 2E** and **2F**), clarifying the distinct response of Tbc1d24 to DON and MPA compared to Impdh. Transient treatment with MPA showed that Tbc1d24 cytoophidia reformed 48 hrs after washing out MPA, revealing the reversible nature of Tbc1d24 cytoophidia formation (**S2 Fig**). Acivicin treatment also resulted in significant suppression of Tbc1d24 cytoophidia (**Fig 2G** and **2H**), making a contrast to Impdh cytoophidia that were markedly induced (**S3C** and **S3D Fig**).

Ctps cytoophidia formation is known to be augmented by DON and Acivicin [22]. Tbc1d24 cytoophidia were decreased significantly by DON (**Fig 2C** and **2D**) and Acivicin (**Fig**

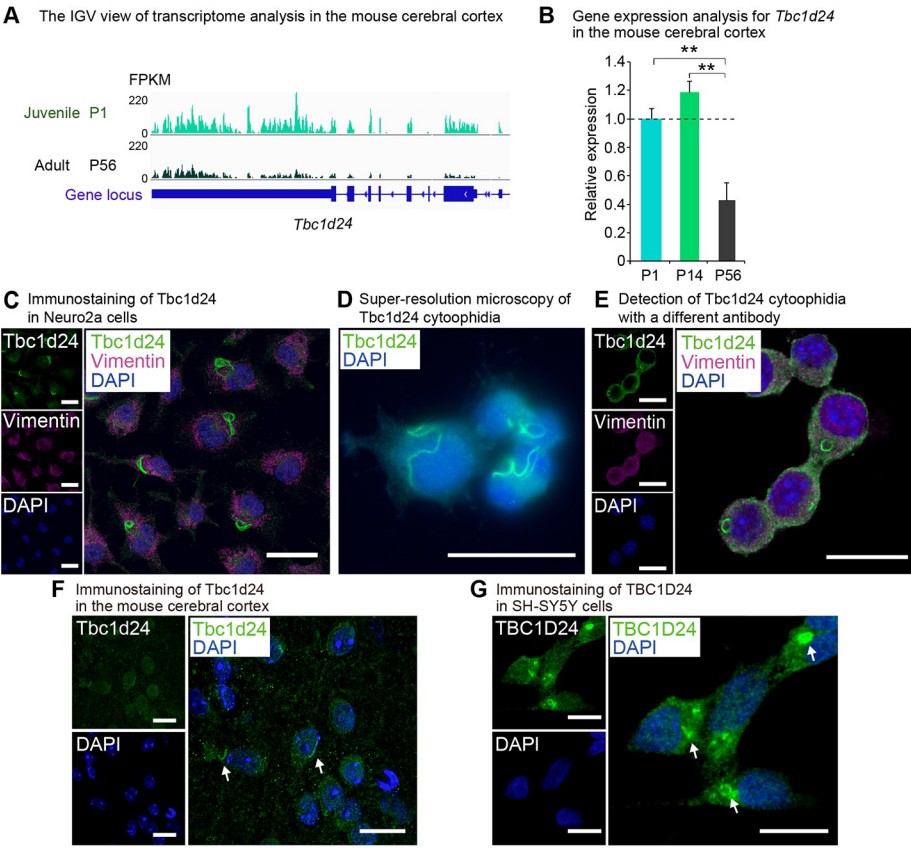

**Fig 1. *Tbc1d24* is expressed predominantly in juvenile in the cerebral cortex and forms cytoophidia.** (A) IGV view of the transcriptome analysis in the mouse cerebral cortex at postnatal day (P) 1 and P56. Numbers next to the read patterns (0 and 220) indicate fragments per kilobase of exon per million reads mapped (FPKM) values. (B) qPCR analysis of mouse *Tbc1d24* in the postnatal mouse cerebral cortex at P1, P14 and P56. Data were normalized to *Polr2a* (n = 3). (C) Immunofluorescence analysis of Tbc1d24 in Neuro2a cells. Cells were also stained with the acetylated tubulin antibody and DAPI. Scale bar = 25 μm. (D) Super-resolution microscopy of Tbc1d24 cytoophidia in Neuro2a cells. Scale bar = 25 μm. (E) Immunofluorescence analysis of Tbc1d24 and Vimentin with a different antibody against Tbc1d24 (LSBio, LS-C679739) in Neuro2a cells. Scale bar = 25 μm. (F) Immunofluorescence analysis of Tbc1d24 in the mouse cortical tissue. The arrows indicate Tbc1d24 cytoophidia. Scale bar = 25 μm. (G) Immunofluorescence analysis of TBC1D24 in SH-SY5Y cells. The arrows indicate TBC1D24 cytoophidia. Scale bar = 10 μm. $^{**}p<0.01$; Student's *t*-test. The data were presented as the means ± standard error of the mean (SEM).

**2G** and **2H**), revealing the distinct responses of Tbc1d24 compared to Ctps. To examine whether Tbc1d24 protein expression was suppressed by these compounds, we performed western blot analysis that showed Tbc1d24 protein was not decreased significantly by these treatments, indicating that the decrease of cytoophidia was not due to suppression of Tbc1d24 protein (**S4 Fig**). These findings indicate Tbc1d24 cytoophidia are distinct from Impdh and Ctps cytoophidia.

## Tbc1d24 cytoophidia is increased by loss of cellular juvenescence

We next investigated physiological relevance of Tbc1d24 cytoophidia. Because Tbc1d24 was expressed in a juvenile cell, we evaluated the role of the cytoophidia for cellular juvenescence. The loss of cellular juvenescence was induced by depletion of *Gm14230*. *Gm14230* is a JALNC and is essential for cellular juvenescence [2]. The siRNA-mediated depletion of *Gm14230* (**Fig 3A**) induced cytoplasmic extensions and impairment of cellular growth (**Fig 3B** and **3C**),

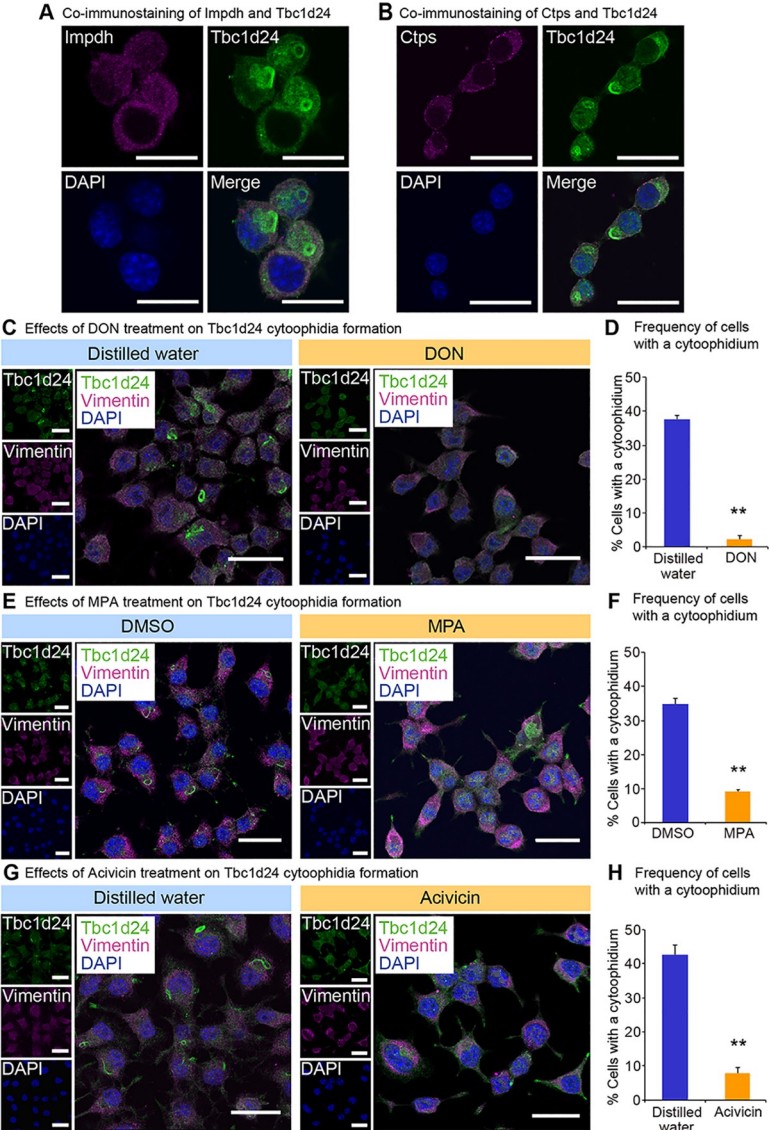

**Fig 2. Tbc1d24 cytoophidia are distinct from Impdh and Ctps cytoophidia.** (A) Costaining of Impdh and Tbc1d24 in Neuro2a cells. Scale bar = 25 μm. (B) Costaining of Ctps and Tbc1d24 in Neuro2a cells. Scale bar = 25 μm. (C) Immunofluorescence analysis of Tbc1d24 in Neuro2a cells treated with control distilled water or 2 mM 6-diazo-5-oxo-L-norleucine (DON) for 24 hours (hrs). Scale bar = 25 μm. (D) Frequency of Tbc1d24 cytoophidium-positive Neuro2a cells treated with 2 mM DON for 24 hrs. (E) Immunofluorescence analysis of Tbc1d24 in Neuro2a cells treated with control dimethyl sulfoxide (DMSO) or 2 μM mycophenolate (MPA) for 24 hrs. Scale bar = 25 μm. (F) Frequency of Tbc1d24 cytoophidium-positive Neuro2a cells treated with 2 μM MPA for 24 hrs. (G) Immunofluorescence analysis of Tbc1d24 in Neuro2a cells treated with control distilled water or 2 mM Acivicin for 24 hrs. Scale bar = 25 μm. (H) Frequency of Tbc1d24 cytoophidium-positive Neuro2a cells treated with 2 mM Acivicin for 24 hrs. $^{**}p<0.01$; Student's *t*-test. The data were presented as the means ± SEM.

hallmarks of juvenescence loss. The *Gm14230*-depleted Neuro2a cells exhibited positivity for senescence-associated beta-galactosidase (SA-β-gal) activity (**Fig 3D** and **3E**). The expression of senescence-associated proteins was enhanced by *Gm14230* depletion, corroborating that *Gm14230* depletion led to the loss of cellular juvenescence (**Fig 3F** and **3G**). The loss of cellular juvenescence resulted in more frequent formation of cytoophidia, revealing that Tbc1d24 cytoophidia formation was inducibly formed by the loss of cellular juvenescence (**Fig 3H** and

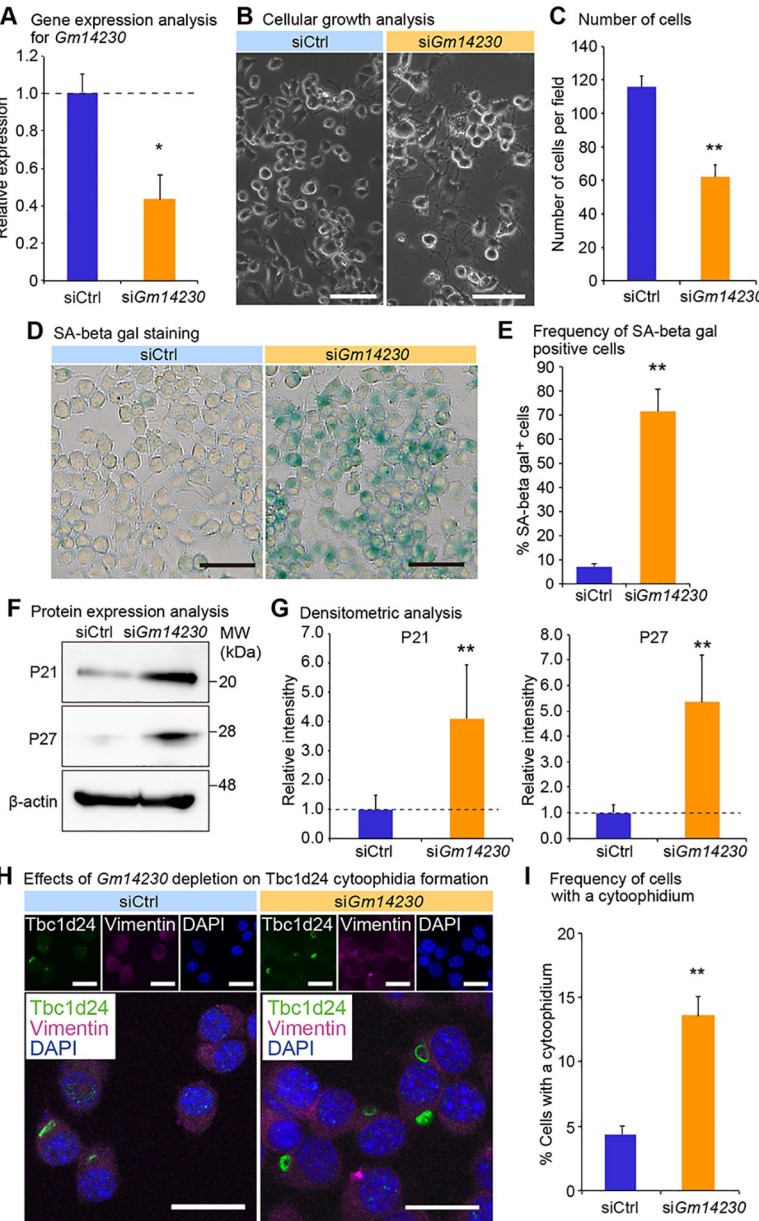

**Fig 3. *Gm14230* controls cellular juvenescence and Tbc1d24 cytoophidia.** (A) qPCR analysis of *Gm14230* in Neuro2a cells transfected with control siRNA (siCtrl) or *Gm14230* siRNA. Data were normalized to the expression of *Tubb5* (n = 3). (B) The appearance of Neuro2a cells 72 hrs after transfection of control siRNA or *Gm14230* siRNA. Scale bar = 100 μm. (C) Number of cells per field 72 hrs after transfection with control siRNA or *Gm14230* siRNA. (D) Senescence-associated (SA) beta-galactosidase activity staining in Neuro2a cells 72 hrs after transfection with control siRNA or *Gm14230* siRNA. Scale bar = 100 μm. (E) Number of SA beta-gal-positive cells per field 72 hrs after transfection with control siRNA or *Gm14230* siRNA. (F) Western blot analysis of Neuro2a cells 72 hrs after transfection with control siRNA or *Gm14230* siRNA. β-actin (Actb) was used as a loading control. MW, molecular weight. (G) The densitometric analysis for western blot analyses from which the representative images were shown as **Fig 3F**. The intensity of the bands was quantified and normalized to those of Actb. The ratios to Actb were further normalized to siCtrl. (H) Immunofluorescence analysis of Tbc1d24 and Vimentin in Neuro2a cells transfected with control siRNA or *Gm14230* siRNA. Scale bar = 25 μm. (I) Frequency of Tbc1d24 cytoophidia in Neuro2a cells transfected with control siRNA or *Gm14230* siRNA. *$p < 0.05$ and **$p < 0.01$; Student's *t*-test. The data were presented as the means ± SEM.

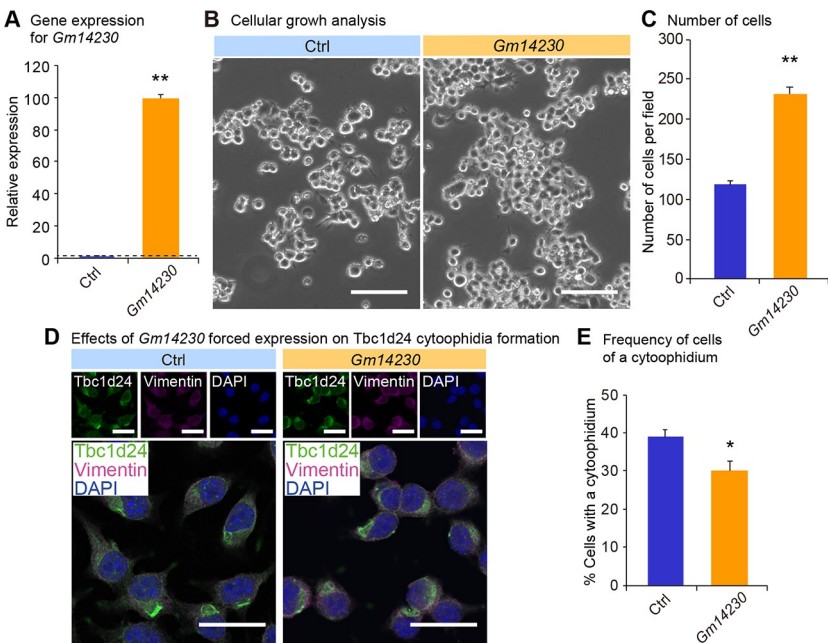

**Fig 4. Influence of *Gm14230* expression on Tbc1d24 cytoophidia formation.** (A) qPCR analysis of *Gm14230* in Neuro2a cells transfected with the empty control or the *Gm14230* plasmid. Data were normalized to *Tubb5* (n = 3). (B) The appearance of Neuro2a cells 72 hrs after transfection of the empty control or the *Gm14230* plasmid. Scale bar = 100 μm. (C) Number of cells per field 72 hrs after transfection of the empty control or the *Gm14230* plasmid. (D) Immunofluorescence analysis of Tbc1d24 and Vimentin in Neuro2a cells transfected with the empty control or the *Gm14230* plasmid. Scale bar = 100 μm. (E) Frequency of Tbc1d24 cytoophidia in Neuro2a cells transfected with the empty control or the *Gm14230* plasmid. $^*p < 0.05$ and $^{**}p < 0.01$; Student's *t*-test. The data were presented as the means ± SEM.

**3I**). Based on the finding that *Gm14230* depletion induces the cytoophidia, we next asked whether the forced expression of *Gm14230* suppressed the cytoophidia formation. Transfection of *Gm14230* (**Fig 4A**) enhanced the growth of Neuro2a cells significantly (**Fig 4B** and **4C**). This enhancement of the cellular growth resulted in the significant suppression of the cytoophidia formation, suggesting the cytoophidia formation was impaired in the rigorously growing cells (**Fig 4D** and **4E**). These findings indicate that Tbc1d24 cytoophidia formation is negatively regulated by *Gm14230*.

Inspired by the finding that cytoophidia formation was induced by the loss of cellular juvenescence, we next asked the biological role of cytoophidia in the juvenescence-losing cells. Forced expression of Tbc1d24 ameliorated the growth impairment caused by *Gm14230* depletion, implying the role of Tbc1d24 to resist the progression of the loss of cellular juvenescence (**S5 Fig**). To further assess the resistive role of Tbc1d24 in the juvenescence loss, we next asked the effect of *Tbc1d24* suppression in the context of *Gm14230* depletion. Simultaneous knockdown of *Gm14230* and *Tbc1d24* resulted in synergistic effects on the cell growth impairment (**S6A** and **S6B Fig**) and cell toxicity (**S6C** and **S6D Fig**), corroborating that Tbc1d24 constitutes a protective machinery against the juvenescence loss.

## Zeocin-induced loss of cellular juvenescence increases Tbc1d24 cytoophidia

To further evaluate the role of cytoophidia in the cellular juvenescence, we utilized the assay system in which cells were treated with a genotoxic agent zeocin to induce the loss of cellular juvenescence [2, 45]. Zeocin treatment provoked cell growth impairment and cytoplasmic

enlargement, suggesting that cells lost cellular juvenescence (**Fig 5A** and **5B**). Tbc1d24 cytoophidia were more frequently observed in zeocin-treated cells compared to control cells (**Fig 5C** and **5D**). Kinetics for zeocin treatment-induced cytoophidia were analyzed by the time course experiment as follows. Longer incubation with zeocin resulted in more obvious cellular shape changes that were accompanied by more frequent (**S7A** and **S7B Fig**) and longer (**S7A** and **S7C Fig**) cytoophidia formation, indicating the positive correlation of the juvenescence loss and cytoophidia formation. The finding of time-dependent increment of the cytoophidia led us to a question whether the Tbc1d24 protein half-life was affected by the cytoophidia formation. The cycloheximide (CHX) chase assay for Tbc1d24 protein showed elongated half-life of Tbc1d24 protein in zeocin-treated cells compared to control cells, suggesting that the enhanced stability of protein contributed to the larger cytoophidia (**S8 Fig**).

To assess a resistive role of Tbc1d24 in zeocin-induced juvenescence loss, we first tested the effect of *Tbc1d24* forced expression. The transfection of *Tbc1d24* alleviated cellular growth impairment and morphological changes induced by zeocin treatment, indicating Tbc1d24 plays a protective role against the cytotoxicity provoked by zeocin-induced juvenescence loss (**S9 Fig**). Furthermore, we evaluated the effect of the depletion of *Tbc1d24* on the zeocin-induced juvenescence loss (**Fig 5E**). The impairment of cellular growth induced by zeocin was exacerbated by *Tbc1d24* depletion, corroborating the cytoprotective function of Tbc1d24 (**Fig 5F** and **5G**). The detection of dead cells by Sytox blue staining revealed higher toxicity with *Tbc1d24*-depleted cells compared to control siRNA-transfected cells, suggesting Tbc1d24 protected cells from the cellular stress given by the zeocin-induced juvenescence loss (**Fig 5H** and **5I**). These findings suggested the cytoprotective role of Tbc1d24 cytoophidia in the zeocin-induced loss of cellular juvenescence.

## Tbc1d24 GAP enzymatic activity is impaired by cytoophidia formation

Lastly, we asked whether the cytoprotective role of Tbc1d24 was a consequence of alteration of GAP enzymatic activity. We interrogated the GAP activity of Tbc1d24 by examining GTP-bound status of Arf6, a substrate of Tbc1d24 [3]. The increased Tbc1d24 cytoophidia induced by *Gm14230* depletion was correlated with impaired GAP activity of Tbc1d24, indicating that Tbc1d24 GAP activity was suppressed by cytoophidia formation (**Fig 6A and 6B**). We next evaluated the correlation of the magnitude of cytoophidia formation and the GAP activity. To assess the correlation between the cytoophidia size and the GAP activity, cells were analyzed for cytoophidia formation and GAP activity in a time course-dependent manner after *Gm14230* depletion. This analysis revealed the time course-dependent impairment of Tbc1d24 GAP activity (**S10A** and **S10B Fig**) that was accompanied with cytoophidia growth in number (**S10C** and **S10D Fig**) and size (**S10C** and **S10E Fig**). These findings suggest that the cytoophidia formation was associated with impairment of Tbc1d24 activity.

Together, our analyses reveal the Tbc1d24 protein forms cytoophidia in neuronal cells both *in vitro* and *in vivo*. The cytoophidia formation is controlled by JALNC *Gm14230* and is induced upon the loss of cellular juvenescence caused by *Gm14230* depletion and zeocin treatment. Depletion of the cytoophidia exacerbates the cell toxicity provoked by loss of the cellular juvenescence. The cytoophidia formation was associated with impaired enzymatic activity of Tbc1d24 (**Fig 6C**).

## Discussion

We in this paper described that Tbc1d24 protein formed cytoophidia in neuronal cells and that Tbc1d24 cytoophidia possessed a protective role against the cell stress provoked by loss of cellular juvenescence. The formation of Tbc1d24 cytoophidia was augmented by the loss of

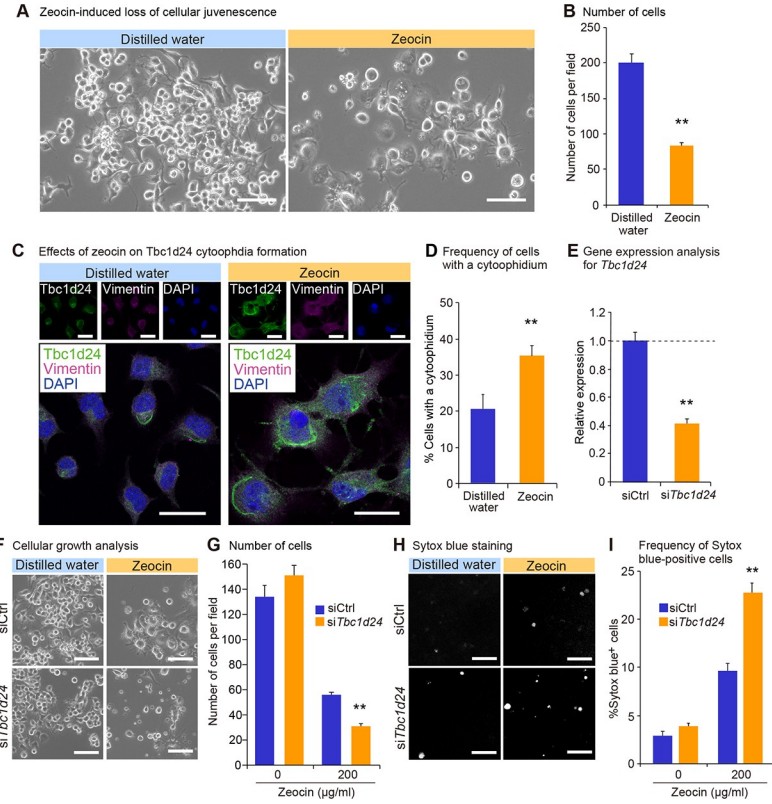

**Fig 5. Tbc1d24 cytoophidia formation is enhanced in zeocin-induced loss of cellular juvenescence.** (A) The appearance of Neuro2a cells 72 hrs after treatment with zeocin. Scale bar = 100 μm. (B) Number of cells per field 72 hrs after treatment with or without zeocin. (C) Immunofluorescence analysis of Tbc1d24 and Vimentin treated with or without zeocin for 72 hrs. Scale bar = 25 μm. (D) Frequency of cytoophidia in Neuro2a cells with or without zeocin treatment for 72 hrs. (E) qPCR analysis of *Tbc1d24* in Neuro2a cells transfected with control siRNA or *Tbc1d24* siRNA. Data were normalized to *Tubb5* (n = 3). (F) The appearance of Neuro2a cells treated with or without zeocin for 96 hrs and simultaneously transfected with control siRNA or *Tbc1d24* siRNA. Scale bar = 100 μm. (G) Number of cells per field 96 hrs after treatment with or without zeocin. (H) The appearance of cells with Sytox blue staining in the same fields as in (F). The gray color dots indicate dead cells stained with Sytox blue. (I) Frequency of Sytox blue-positive cells. **$p < 0.01$; Student's *t*-test. The data were presented as the means ± SEM.

cellular juvenescence induced by depletion of JALNC *Gm14230* and treatment with zeocin, a genotoxic agent. The cytoprotective function of Tbc1d24 was revealed by synergistic effect of *Tbc1d24* depletion on the cell toxicity caused by *Gm14230* depletion and zeocin treatment. The GAP activity of Tbc1d24 was impaired by cytoophidia formation, indicating that cytoophidia formation was associated with suppressed Tbc1d24 activity. Thus, Tbc1d24 constitutes the cell stress-coping machinery under the control of *Gm14230*.

The cytoprotective property of Tbc1d24 cytoophidia was demonstrated by the experiment in which cells depleted of *Tbc1d24* were more susceptible to zeocin toxicity. Loss of cellular juvenescence caused by *Gm14230* depletion or zeocin treatment significantly increased cytoophidia formation, indicating cytoophidia formation was suppressed in juvenile cells. To assess the physiological function of Tbc1d24 cytoophidia, the cell viability was tested in *Tbc1d24*-depleted cells after zeocin treatment. More dead cells were found among *Tbc1d24*-depleted cells compared to control siRNA-transfected cells, suggesting a protective role of *Tbc1d24* when cells were induced to lose cellular juvenescence. *Gm14230* may function as a stress sensor that mediates the stress signals to the cytoophidia formation. As the protein that is highly

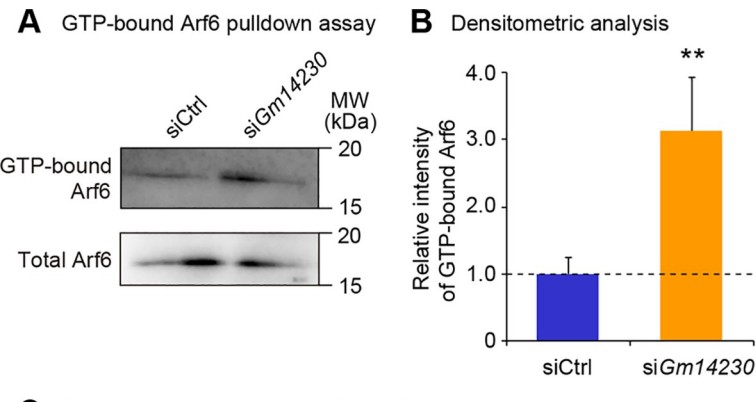

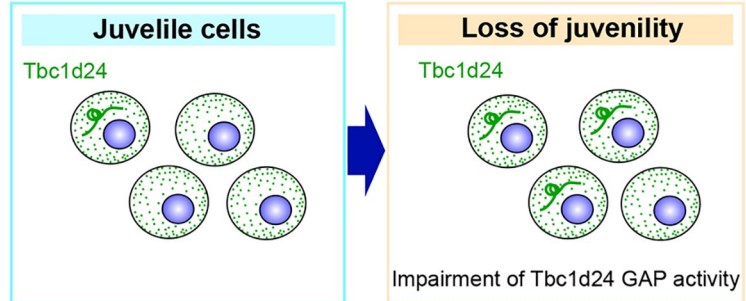

**Fig 6. Lower GAP activity of Tbc1d24 is associated with cytoophidia formation.** (A) GTP-bound Arf6 pulldown assay in Neuro2a cells transfected with control siRNA or *Gm14230* siRNA. Total Arf6 levels were examined to evaluate the equal expression of Arf6 protein after *Gm14230*-depletion. (B) The densitometric analysis for western blot analyses from GTP bound Arf6 pulldown assay. The intensity of the bands was quantified and normalized to those of total Arf6. The ratios to total Arf6 were further normalized to siCtrl. $^{**}p < 0.01$; Student's *t*-test. Data were represented as the means ± SEM. (C) Schematics for the role of Tbc1d24 cytoophidia to control GAP activity in a cellular juvenescence-dependent manner.

expressed in juvenile cells, Tbc1d24 may function to maintain cellular juvenescence by forming the cytoophidia.

As above, Tbc1d24 cytoophidia may function as the stress-response element and protect cells from zeocin treatment- and *Gm14230*-depletion-induced loss of the cellular juvenescence. This idea was further supported by the findings that *Tbc1d24*-depletion increased cellular toxicity upon *Gm14230* loss (**S6 Fig**) and that *TBC1D24* forced expression rescued the cell toxicity of zeocin treatment (**S9 Fig**).

The filamentous macrostructure of Tbc1d24 cytoophidia were similar to that of Impdh and Ctps cytoophidia, but we demonstrated the Tbc1d24 cytoophidia were distinct from the Impdh and Ctps cytoophidia by a series of experiments. Treatment with DON, MPA and Acivicin suppressed Tbc1d24 cytoophidia formation without affecting Tbc1d24 protein levels. These compounds decrease intracellular GTP through Impdh inhibition. The lower GTP levels may affect cytoophidia formation through modulating the Tbc1d24 enzymatic activity. Further experiments are required to assess the association with intracellular GTP levels and Tbc1d24 molecular behaviors.

We consider that Tbc1d24 cytoophidia are distinct from mere protein aggregations and function as the reservoir of the enzyme to cope with cellular stresses. Our work suggests that Tbc1d24 cytoophidia function as a reservoir of Tbc1d24 protein to adjust GAP activity. Ctps cytoophidia were reported to correlate with both negative [27, 46] and positive [47] activities.

Impdh cytoophidia were correlated with positive activity [41]. Our data show a negative correlation between Tbc1d24 cytoophidia formation and GAP activity. This finding suggests that Tbc1d24 cytoophidia functions as a Tbc1d24 reservoir to store Tbc1d24 in an inactive form in the context of the cellular stress response.

Mutations in *TBC1D24* have been discovered in patients with EE, a severe brain syndrome that develops in juvenile in humans. Patients with EE exhibit cognitive, behavioral and neurological deficits. The exact genotype-phenotype correlation is under investigation [48]. Most of the affected individuals possess homozygous mutations in TBC domain or compound heterozygous mutations in TBC or TLDc domains. Our findings imply a therapeutic potential of targeting Tbc1d24 cytoophidia in neuronal cells for the treatment of EE. Mutations in *TBC1D24* may affect cytoophidia formation, compromising the cellular stress response. It might be possible to induce TBC1D24 cytoophidia via a compound to cope better with the cellular stresses, which may lead to a new therapeutic approach for the detrimental neurological disorders.

We revealed the property of Tbc1d24 protein to form cytoophidia in response to loss of cellular juvenescence. Tbc1d24 cytoophidia formation was augmented by depletion of JALNC *Gm14230* and the genotoxic agent zeocin. Depletion of Tbc1d24 cytoophidia enhanced cellular toxicity against zeocin treatment, indicating the cytoprotective role for Tbc1d24 cytoophidia. Cytoophidia formation correlated with impairment of GAP activity, thus implying that Tbc1d24 cytoophidia function as the repressive machinery for Tbc1d24 enzymes. Collectively, our findings illustrate the new role of Tbc1d24 for cytoophidia formation and stress response, illuminating a novel therapeutic approach for the neurological disorders.

## Supporting information

**S1 Fig. Co-immunoprecipitation analysis of Tbc1d24 with Ctps or Impdh.** Co-immunoprecipitation assay performed in Neuro2a cells overexpressing FLAG-*TBC1D24*. The pulldown reactions were performed with anti-FLAG or control rabbit antibody. The precipitated protein samples were analyzed by SDS-PAGE followed by immunoblot analysis for Ctps, Impdh and FLAG. IP, immunoprecipitation.
(PDF)

**S2 Fig. Reversibility of the Tbc1d24 cytoophidia formation.** (A) Immunofluorescence analysis of Tbc1d24 in Neuro2a cells treated with 2 μM MPA or control DMSO for 24 hrs followed by washout of MPA. After removing MPA, cells were further cultured for 24 or 48 hrs prior to Tbc1d24 immunostaining. DAPI was used to stain nuclei. Scale bar = 25 μm. (B) Frequency of cells with a cytoophidium. n.s., not significant. $^{**}p < 0.01$; Student's *t*-test. The data were presented as the means ± SEM.
(PDF)

**S3 Fig. The Tbc1d24 cytoophidium is distinct from Impdh or Ctps cytoophidium.** (A) Tbc1d24 immunocytochemistry in Neuro2a cells treated with 2 mM Acivicin or control distilled water for 24 hrs. DAPI was used to stain nuclei. Scale bar = 25 μm. (B) Frequency of Tbc1d24 cytoophidia in Neuro2a cells treated with Acivicin or distilled water. (C) Impdh immunocytochemistry in Neuro2a cells treated with Acivicin or distilled water for 24 hrs. DAPI was used to stain nuclei. Scale bar = 25 μm. (D) Frequency of Impdh cytoophidia in Neuro2a cells treated with Acivicin or distilled water. (E) Ctps immunocytochemistry in Neuro2a cells treated with Acivicin or distilled water for 24 hrs. DAPI was used to stain nuclei. Scale bar = 25 μm. (F) Frequency of Ctps cytoophidia in Neuro2a cells treated with Acivicin or distilled water. n.d., not detected, $^{**}p < 0.01$; Student's *t*-test. The data were presented as the

means ± SEM.
(PDF)

**S4 Fig. Tbc1d24 protein levels did not decrease significantly with the treatment with DON, Acivicin or MPA.** (A) Western blot analysis of Neuro2a cells treated with control distilled water, 2 mM DON or 2 mM Acivicin for 24 hrs. Actb was used as loading control. (B) The densitometric analysis for the western blot analysis of Tbc1d24 protein levels in Neuro2a cells treated with DON or Acivicin. The intensity of the bands was quantified and normalized to Actb. The ratios were further normalized to Ctrl treatment. n.s., not significant, Student's *t*-test. The data were presented as the means ± SEM. (C) Western blot analysis of Neuro2a cells treated with 2 μM MPA or DMSO for 24 hrs. Actb was used as loading control. (D) The densitometric analysis for the western blot analysis of Tbc1d24 protein levels in Neuro2a cells treated with MPA. The intensity of the bands was quantified and normalized to Actb. The ratios were further normalized to Ctrl treatment. n.s., not significant, Student's *t*-test. The data were presented as the means ± SEM.
(PDF)

**S5 Fig. Forced expression of *Tbc1d24* exerts the protective effect in the loss of cellular juvenescence.** (A) The effect of *Tbc1d24* overexpression was examined in si*Gm14230*-induced loss of juvenescence. The appearance of Neuro2a cells transfected with control empty plasmid or T*BC1D24* plasmid simultaneously with control siRNA or *Gm14230* siRNA. Scale bar = 100 μm. (B) Number of cells per field. The growth was investigated in Neuro2a cells transfected with or control empty plasmid or *TBC1D24* plasmid simultaneously with control siRNA or *Gm14230* siRNA. $^{**}p < 0.01$; Student's *t*-test. The data were presented as the means ± SEM.
(PDF)

**S6 Fig. The protective effect of Tbc1d24 in the *Gm14230*-depletion-induced loss of cellular juvenescence.** (A) Appearance of Neuro2a cells 48 hrs after the simultaneous knockdown of *Gm14230* and *Tbc1d24*. Indicated siRNAs were transfected in Neuro2a cells for the evaluation of cell growth. Scale bar = 100 μm. (B) Quantification of Neuro2a cells transfected the indicated siRNAs. (C) Sytox blue staining in the same fields as in (A). The gray dots indicated dead cells stained by Sytox blue. (D) Frequency of Sytox blue-positive cells. $^{**}p < 0.01$; Student's *t*-test. The data were presented as the means ± SEM.
(PDF)

**S7 Fig. Tbc1d24 cytoophidia formation is promoted by zeocin induced-cellular senescence in the time-dependent manner.** (A) Tbc1d24 immunocytochemistry in Neuro2a cells treated with zeocin for 0, 48, 72 and 96 hrs. DAPI was used to stain nuclei. Scale bar = 25 μm. (B) Frequency of cells positive for Tbc1d24 cytoophidium in Neuro2a cells treated with zeocin for 0, 48, 72 and 96 hrs. n.s. not significant. $^{**}p < 0.01$; Student's *t*-test. The data were presented as the means ± SEM. (C) Length (μm) of Tbc1d24 cytoophidium in Neuro2a cells treated with zeocin for 0, 48, 72 and 96 hrs. $^{**}p < 0.01$; Student's *t*-test. The data were presented as the means ± SEM.
(PDF)

**S8 Fig. Tbc1d24 protein is stabilized by cytoophidia formation.** (A) The half-life of Tbc1d24 protein was analyzed by the cycloheximide (CHX) chase assay in Neuro2a cells treated with control distilled water or zeocin for 72 hrs. The cells were treated with CHX 20 μg/ml for 0, 4, 8 and 12 hrs. Actb was used as a loading control. (B) Quantitative analysis of western blots for Tbc1d24 was shown. The intensity of the bands was quantified and normalized to those of

Actb. The ratios to Actb were further normalized to 0 hr. $^{*}p < 0.05$; Student's *t*-test. The data were presented as the means ± SEM.
(PDF)

**S9 Fig. Forced expression of *Tbc1d24* exerts the protective effect in zeocin-induced cellular senescence.** (A) The effect of *Tbc1d24* overexpression was examined in zeocin-induced cellular senescence. Prior to zeocin treatment, Neuro2a cells were transfected with *TBC1D24* or control empty plasmid. The cell appearance was observed to evaluate cell viability. Scale bar = 100 μm. (B) The number of cells per field. The growth was investigated in Neuro2a cells transfected with *TBC1D24* or control empty plasmid prior to zeocin treatment. $^{**}p < 0.01$; Student's *t*-test. The data were presented as the means ± SEM.
(PDF)

**S10 Fig. Correlation of the cytoophidia size and the GAP activity of Tbc1d24.** (A) GTP-bound Arf6 pulldown assay in Neuro2a cells at 0, 48 and 72 hrs after transfection with *Gm14230* siRNA. Total Arf6 levels were examined as loading control. (B) The densitometric analysis of GTP-bound Arf6 pulldown assays. The intensity of the bands was quantified and normalized to those of total Arf6. The ratios to total Arf6 were further normalized to 0 hr. $^{*}p < 0.05$ and $^{**}p < 0.01$; Student's *t*-test. The data were presented as the means ± SEM. (C) Immunofluorescence analysis of Tbc1d24 in Neuro2a cells at 0, 48 and 72 hrs after transfection with *Gm14230* siRNA. DAPI was used to stain nuclei. Scale bar = 25 μm. (D) Frequency of Tbc1d24 cytoophidia positive cells at 0, 48 and 72 hrs after transfection with *Gm14230* siRNA. $^{*}p < 0.05$ and $^{**}p < 0.01$; Student's *t*-test. The data were presented as the means ± SEM. (E) Length (μm) of Tbc1d24 cytoophidium in Neuro2a cells was measured at 0, 48 and 72 hrs after transfection with *Gm14230* siRNA. $^{*}p < 0.05$ and $^{**}p < 0.01$; Student's *t*-test. The data were presented as the means ± SEM.
(PDF)

**S1 Raw images. The uncropped images of western blot.**
(PDF)

**S1 Table. Expression levels of the relevant genes.** The expression levels of the relevant genes in this study. Shown were the fragments per kilobase of transcript per million fragments sequenced (FPKM) values obtained from the RNA-seq analysis of the mouse cerebral cortex at postnatal day 1 (P1) and P56.
(PDF)

**S1 File.**
(XLSX)

## Acknowledgments

We would like to thank all the researchers of National Cerebral and Cardiovascular Center Research Institute for their helpful discussions and sincere cooperation. This study was supported by the Central Research Laboratory at Shiga University of Medical Science.

## Author Contributions

**Conceptualization:** Takao Morimune, Masaki Mori.

**Data curation:** Takao Morimune, Masaki Mori.

**Formal analysis:** Takao Morimune, Masaki Mori.

**Funding acquisition:** Takao Morimune, Masaki Mori.

**Investigation:** Takao Morimune, Ayami Tano, Yuya Tanaka, Haruka Yukiue, Takefumi Yamamoto, Yoshihiro Maruo, Masaki Mori.

**Methodology:** Takao Morimune, Masaki Mori.

**Project administration:** Masaki Mori.

**Resources:** Takao Morimune, Masaki Mori.

**Software:** Takao Morimune, Masaki Mori.

**Supervision:** Yoshihiro Maruo, Masaki Nishimura, Masaki Mori.

**Validation:** Takao Morimune, Masaki Mori.

**Visualization:** Takao Morimune, Masaki Mori.

**Writing – original draft:** Takao Morimune, Masaki Mori.

**Writing – review & editing:** Takao Morimune, Ikuo Tooyama, Yoshihiro Maruo, Masaki Nishimura, Masaki Mori.

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
