## [Decision Letter · Decision Letter 0]

6 Oct 2020

PONE-D-20-24106

Gm14230 controls Tbc1d24-cytoophidia and neuronal cellular juvenescence.

PLOS ONE

Dear Dr. Dr. Mori,

Thank you for submitting your manuscript to PLOS ONE. After careful consideration, we feel that it has merit but does not fully meet PLOS ONE’s publication criteria as it currently stands. Therefore, we invite you to submit a revised version of the manuscript that addresses the points raised during the review process.

Please submit your revised manuscript within 3 months of this decision. If you will need more time than this to complete your revisions, please reply to this message or contact the journal office at plosone@plos.org. Please include the following items when submitting your revised manuscript:

We look forward to receiving your revised manuscript.

Kind regards,

Krishna M. Bhat, M.D., Ph.D.

Academic Editor

PLOS ONE

Journal Requirements:

Reviewers' comments:

Reviewer's Responses to Questions

**Comments to the Author**

1. Is the manuscript technically sound, and do the data support the conclusions?

Reviewer #1: Yes

Reviewer #2: Partly

2. Has the statistical analysis been performed appropriately and rigorously? 

Reviewer #1: Yes

Reviewer #2: No

3. Have the authors made all data underlying the findings in their manuscript fully available?

Reviewer #1: Yes

Reviewer #2: No

4. Is the manuscript presented in an intelligible fashion and written in standard English?

Reviewer #1: Yes

Reviewer #2: Yes

5. Review Comments to the Author

Reviewer #1: The manuscript entitled "Gm14230 controls Tbc1d24-cytoophidia and neuronal cellular juvenescence" by Morimune et al. studied the formation of Tbc1d24 cytoophidia and its association with Tbc1d24 enzymatic activity and cellular juvenescence. They found in their in vivo and in vitro studies: Tbc1d24 forms cytoophidia in response to the loss of cellular juvenescence caused by depletion of Gm14230, which could serve as a regulatory machinery for Tbc1d24 enzymatic activity. The Tbc1d24 cytoophidia was also found to be quite different from Ctps and Impdh cytoophidia. Overall, the paper is well-written, and the quality of the presented research work is good. Although I think that it is a good candidate for being published in PLOS one, the following questions should be addressed properly.

1. Whether these cytoophidia proteins interact with each other biochemically? IFS only represents the distribution of the major portion of the proteins, and there is still a possibility of co-existence of the cytoophidia proteins.

2. Is cytoophidia protective or toxic to cell survival?

3. What is the difference between cytoophidia and other aggregations?

4. Is cytoophidia formation reversible when cell meets stress?

5. Are these cytoophidia the mixture of different proteins? Can we use Co-IP experiments to prove it?

6. Is there any correlation between the size of Cytoophidia aggregates or their growth rate and the degree of Tbc1d24 enzymatic activity impairment?

7. Is there any correlation between the size of Cytoophidia aggregates and the degree of cellular senescence?

Specific questions:

1. There is no figure as 1E.

2. What are the MPA, DON, and Acivicin? Authors should include some background information.

3. If we overexpress Tbc1d24 in cells, what will happen? Does it increase the resistance to cellular senescence and protect the cells from external stress?

4. There are no caption and figure legend for figure 1 H.

Reviewer #2: This is a truly exciting paper with a number of provocative conclusions. However, there are a number of concerns that temper enthusiasm.

Some annoying aspects of the paper that impairs the review.

Figure legends are split into multiple locations within the text of the paper.

The supplement figures are not very large and can be easily incorporated within the figures, particularly for Fig S2.

No statistics are provided for the Western blot studies.

The first is that the paper fails to describe the gene changes in the Juvenility-linked transcriptome assay as a table. This table can be in the form of supplementary data.

The cytoophidia findings in Fig 2 raise several questions. Why is there a strong drop in Tbc1d24 cytoophidia with DON, MPA, Acivicin. Is this reversible. What happens to levels of Tbc1d24 protein with these treatments? It is useful to show western blots or other protein quantification methods given that this protein is accumulated while the RNA appears to be dropping.

It is important to demonstrate that the treatments can actually induce cytoophidia with impdh or ctps under these conditions.

An age-dependent time course of Tbc1d24 accumulation as filaments will be useful.

For Fig 5, it will be useful to show the levels of siGm14230 in addition to Tbc1D24 for zeocin treatment.

Finally, it is useful to show the half-life of these proteins with formation of cytoophidia.

6. PLOS authors have the option to publish the peer review history of their article (what does this mean?). If published, this will include your full peer review and any attached files.

Reviewer #1: No

Reviewer #2: **Yes: **Kumar Sambamurti

---

## [Author Response · Author response to Decision Letter 0]

5 Feb 2021

We would like to thank the Editor and the Reviewers for this precious opportunity to revise our manuscript. We made our best efforts to improve the paper based on the Reviewers' suggestions. We appreciate if you find our improvements in the "Response to Reviewers" file.

---

## [Decision Letter · Decision Letter 1]

1 Mar 2021

Gm14230 controls Tbc1d24 cytoophidia and neuronal cellular juvenescence.

PONE-D-20-24106R1

Dear Dr. Mori

The reviewers found your revised manuscript acceptable for publication. Therefore, we’re pleased to inform you that your manuscript has been judged scientifically suitable for publication and will be formally accepted for publication once it meets all outstanding technical requirements.

Kind regards,

Krishna M. Bhat, M.D., Ph.D.

Academic Editor

PLOS ONE

Additional Editor Comments (optional):

Reviewers' comments:

Reviewer's Responses to Questions

**Comments to the Author**

1. If the authors have adequately addressed your comments raised in a previous round of review and you feel that this manuscript is now acceptable for publication, you may indicate that here to bypass the “Comments to the Author” section, enter your conflict of interest statement in the “Confidential to Editor” section, and submit your "Accept" recommendation.

Reviewer #1: All comments have been addressed

2. Is the manuscript technically sound, and do the data support the conclusions?

Reviewer #1: Yes

3. Has the statistical analysis been performed appropriately and rigorously? 

Reviewer #1: Yes

4. Have the authors made all data underlying the findings in their manuscript fully available?

Reviewer #1: Yes

5. Is the manuscript presented in an intelligible fashion and written in standard English?

Reviewer #1: Yes

6. Review Comments to the Author

Reviewer #1: The revised manuscript has addressed most of my questions. I thus recommend to accept it for publication.

7. PLOS authors have the option to publish the peer review history of their article (what does this mean?). If published, this will include your full peer review and any attached files.

Reviewer #1: No

---

## [Editor Report · Acceptance letter]

6 Apr 2021

PONE-D-20-24106R1 

*Gm14230* controls Tbc1d24 cytoophidia and neuronal cellular juvenescence. 

Dear Dr. Mori:

I'm pleased to inform you that your manuscript has been deemed suitable for publication in PLOS ONE. Congratulations! Your manuscript is now with our production department. 

Kind regards, 

on behalf of

Professor Krishna M. Bhat 

Academic Editor

PLOS ONE